

# A web-GIS database of the scientific articles on earthquake-triggered landslides

Luca Schilirò[1], Mauro Rossi[2], Federica Polpetta[1], Federica Fiorucci[2], Carolina Fortunato[1], Paola Reichenbach[2]

[1]CNR IGAG, Area della Ricerca di Roma 1, Strada Provinciale 35d, 9, 00010 Montelibretti (Roma), Italy
[2]CNR IRPI, via Madonna Alta 126, 06128 Perugia, Italy

*Correspondence to*: Luca Schilirò (luca.schiliro@cnr.it)

**Abstract.** Over the last two decades, the topic of earthquake-triggered landslides (EQTLs) has shown an increasing relevance

in the scientific community. This interest is confirmed by the numerous articles published in the international, peer-reviewed journals. In this work we present a database containing a selection of articles published on this topic from 1984 to 2021. The articles were selected through a systematic search on the Clarivate analytics' Web of Science-Core Collection™ online platform and were catalogued into a web-GIS, which was specifically designed to show different types of information. After a general analysis of the database, for each article were identified the bibliometric information (e.g., author(s), title, publication

year), the relevant topic/sub-topic category(s) and the addressed earthquake(s). The analysis allowed to infer general information and statistics on EQTLs (e.g., relevant methodological approaches over time and in relation to the scale of investigation, most-studied events), which can be useful to get a spatial distribution of the articles and a general overview of the topic.

## 1 Introduction

Earthquakes are one of the most threatening and devastating natural hazards worldwide, which cause significant losses of life and damages to human structures and infrastructures. Although most of the damage is related to the partial or full collapse of buildings caused by ground shaking, in many cases other ground effects (e.g., tsunamis, liquefaction, landslides) can significantly increase the impact of the seismic event (e.g., Jibson and Harp, 2012). In particular, earthquake-triggered/coseismic landslides (EQTLs) are responsible for approximately 70% of all earthquake-related fatalities not directly

caused by ground shaking (Marano et al., 2010). Casualties caused by EQTLs are generally related to the collapse of buildings induced by the down-slope movements, which frequently tends to cover up previous structural damage due to the seismic shaking (Bird and Bommer, 2004). EQTLs often interrupt road networks and other transportation infrastructures, hampering rescue, supply, and recovery activities (Allstadt et al., 2022). EQTLs can also partly or completely block river channels, inducing downstream floods (e.g., Fan et al., 2012) and, in the long-term, causing changes of the drainage-basin characteristics

(Keefer, 1999). For all these reasons, EQTLs are an important component of natural hazard assessment in seismically active





areas, and in recent years the study of collateral seismic hazards is becoming an issue of increasing relevance (Wasowski et al., 2011), also in the framework of multi-risk assessments, accounting for potential cascading effects.

In this article we describe the results of a systematic search of scientific papers concerning EQTLs, published in peer-reviewed international journals in the past decades. Searching on the Clarivate analytics' Web of Science-Core Collection™ online

platform, we have collected 804 articles which were organized in a specific web-GIS database for the analysis. The identified articles deal with EQTLs theme from numerous points of view, focusing on a wide range of aspects, such as spatial distribution in relation to earthquake location, triggering and propagation mechanisms, impact on human activities. The aim of this work is to provide a tool which shows the geographical distribution of the articles published on this topic classified using different types of information, which may represent a starting point for future analyses. The manuscript is organized as follows: section

2 describes the structure of the database; Section 3 discusses the type of information addressed by the articles and section 4 summarizes some main conclusions.

## 2 Data and structure of the literature database

The EQTLs literature database was implemented starting from the identification of the articles published in international, peer-reviewed journals. The literature search was performed using Clarivate analytics' Web of Science-Core Collection™ online

platform, which is made up of more than 21,000 peer-reviewed journals worldwide (Web of Science Group, 2021). As search parameters, we used a combination of significant keywords (e.g., *landslide, slope failure, earthquake, seismic, triggered, induced*) and Boolean search criteria applied to the "title", "abstract", and "keywords" of the articles (Reichenbach et al., 2018). A total of 804 articles published from April 1984 to February 2021 were collected and organized in the database. Each publication was verified to check its relevance for the topic of EQTLs. In most cases, it was sufficient to read the title and the

abstract to verify whether an article was relevant for our purpose, but to prepare the final version of the database it was required a double-check of the manuscripts to avoid potential mistakes.

### 2.1 Data categories

Based on our experience and after a preliminary reading of approximately 10% of the articles, we identified a set of information that we consider significant for the analysis. We selected 3 major data categories (i.e., article information, article topic and

earthquake information) further divided into different sub-categories (Table 1).




| Category | | Sub-category | Counts |
|---|---|---|---|
| ID # | A1 | Article identification number | 804 |
| Article information | B1 | Publication year | 1984-2021 |
| | B2 | Author(s) | 1 to 35 per article |
| | B3 | Title | |
| | B4 | Journal | 159 |
| Article topic | C1 | Main topic | 4 |
| | C2 | Sub-topic | 11 |
| Earthquake information | D1 | Earthquake ID | 135 |
| | D2 | Earthquake name | |
| | D3 | Earthquake time | |
| | D4 | Earthquake country | 35 |
| | D5 | Earthquake event type (single\|multiple\|unknown) | 95\|21\|19 |
| | D6 | Earthquake focal mechanism (normal\|normal-strike-slip\|reverse\|reverse-strike-slip\|strike-slip\|strike-slip-normal\|strike-slip-reverse\|unknown) | 16\|4\|41\|11\|25\|9\|10\|19 |
| | D7 | Moment tensor solution | 96 |
| | D8 | Reference for focal mechanism | 23 |
| | D9 | Richter local magnitude | |
| | D10 | Richter surface-wave magnitude | |
| | D11 | Moment magnitude | |
| | D12 | MCS magnitude | |
| | D13 | Epicentre latitude | |
| | D14 | Epicentre longitude | |
| | D15 | Hypocentre depth | |
| | D16 | Earthquake catalogue (ANSS\|NCEI/WDS\|no data) | 117\|15\|3 |

**Table 1: Summary statistics of the categories and sub-categories used in the database. The column *Counts* provides the number of occurrences as given by the authors, when applicable.**




### 2.1.1 Article information

In the database, the information defined for each article are the following: publication year (B1); author(s) (B2); article title (B3) and journal name (B4). The first article of our search was published in April 1984 by Keefer and only 19 articles were published on the topic until 1999 (Figure 1). An increasing trend of the number of published articles can be noted in the following years, particularly from the late 2000s, with a peak in 2019 and 2020 (98 and 97 articles, respectively).

The 804 identified papers were published in 159 different peer-reviewed journals. The *Geological Society of America Bulletin* has published the least number of articles but for the longest period, considering that the first and last article were published in 1984 and 2020, respectively (Figure 2). *Engineering Geology*, *Landslides* and *Geomorphology* have published the largest number of articles (i.e., 231, which represents 28.7% of the total), covering a period from mid 1990s-early 2000s until today.

*Natural Hazards*, *Journal of Mountain Science* and *Bulletin of the Engineering Geology and the Environment* have published 104 papers (12.9% of the total), while *Natural Hazards and Earth System Sciences*, *Bulletin of the Seismological Society of America*, *Environmental Earth Sciences*, *Soil Dynamics and Earthquake Engineering* and *Geosciences* have published 118 papers (14.7% of the total). It is worth noting that *Geosciences* has published the largest annual number of papers with 14 articles in a little more than three years (2018-February 2021).

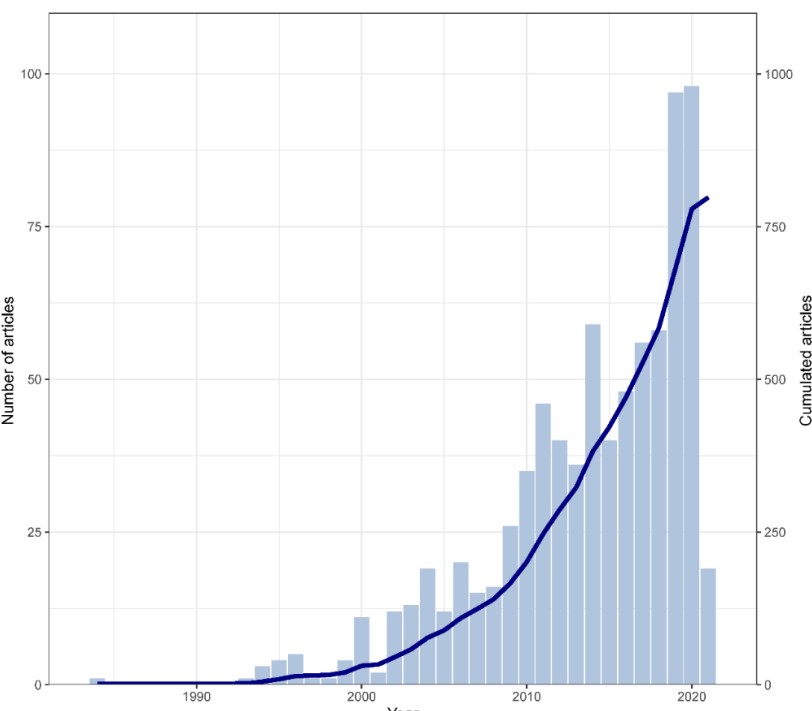

**Figure 1: Analysis of the literature database listing 804 articles in the 37-year period from April 1984 to February 2021. Source of the articles search was the "Web of Science™" (formerly a Thomson Reuters™ product, now part of Clarivate™ Analytics). The graph shows the number of articles per year (vertical blue bars, left y-axis) and their cumulated number (solid blue line, right y-axis).**


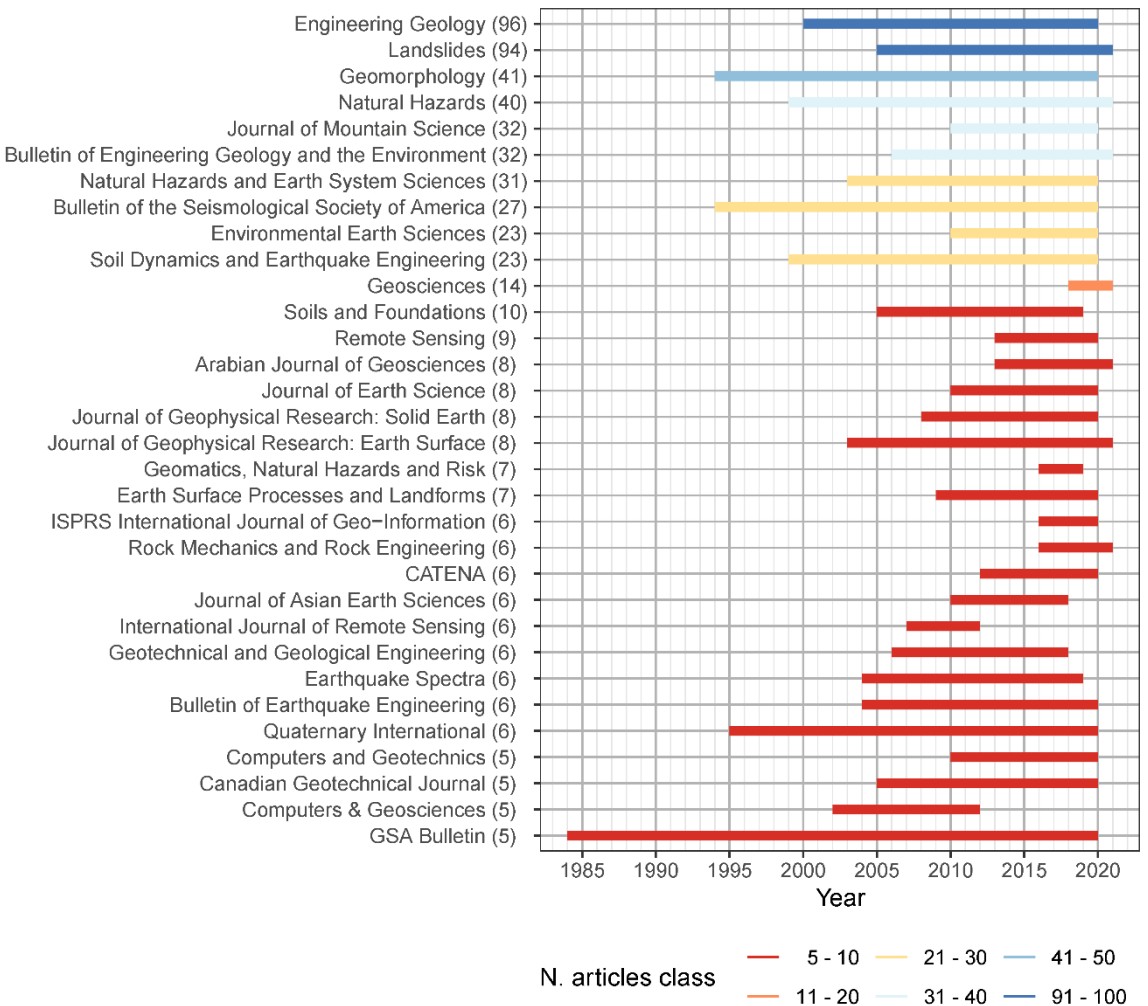

**Figure 2: Ranking of top journals in terms of number of articles published on EQTLs, according to the literature database. The number of published articles is reported in brackets next to the journal name. The colour of horizontal bars indicates the number of articles in six classes.**


Overall, 56.3% of the articles in the literature database were published in the above-mentioned eleven journals, whose scopes include the analysis of natural hazards, geological/geotechnical engineering, and geomorphology. Among these journals, only two (*Bulletin of the Seismological Society of America* and *Soil Dynamics and Earthquake Engineering*) specifically deal with seismology and earthquake engineering.

The analysis of the database revealed that the average number of authors for each paper varied over the years. Excluding the
first article by Keefer (1984), the papers published until 2000 were written, on average, by 2-3 authors. In more recent years, the number increased to 4-5 authors, which suggests the need for different expertise to perform the wide range of research activities related to EQTLs.



### 2.1.2 Article topic

The articles collected in the database were classified in four categories considering the following main topics:

- *Regional landslide analysis*: articles focused on the analysis of EQTLs over large areas.
- *Single landslide analysis*: articles focused on the analysis of single mass movements.
- *Other*: articles presenting various types of research activities related to EQTLs, such as data presentation (e.g., Fortunato et al., 2012; Tanyas et al., 2017; Villani et al., 2018; Rodríguez-Peces et al., 2020), synthesis of historical information and review articles (e.g., Keefer, 1984; 2002; Bird and Bommer, 2004; Wasowski et al., 2011; Fan et al., 2019).

- *Not applicable*: articles focused on topics only partially related to EQTLs (see Sect. 3.1.3).

The articles included in the "regional" and "single landslide analysis" categories were further subdivided in sub-topics that characterize different aspects of the main research activity described in the manuscript (such as mapping, characterization, modelling etc.). In many cases more than one sub-topic has been attributed to a single article. For the "regional landslide analysis" the sub-topics can be summarized as follows:

1) *Regional mapping*: recognition of the spatial distribution of EQTLs.
2) *Regional landslide descriptive statistics*: statistical analysis of the main physical and geometrical features of EQTLs (e.g., number, extension, volume, run-out, type of movement, involved material).
3) *Regional susceptibility/hazard assessment*: modelling and zonation of susceptibility/hazard scenarios for EQTLs.
4) *Regional risk modelling*: assessment of the effects on human activities of EQTLs occurring over large areas.

5) *Regional landslides comparison*: analysis focused on the comparison between different landslide inventory maps, with the aim of evaluating general rules about earthquake induced landslide occurrence.

For the "single landslide analysis", the sub-topics are:

1) *Single failure mapping*: reconstruction of the surficial geometry.
2) *Single failure geotechnical characterization*: definition and description of the geotechnical parameters.

3) *Single failure geophysical characterization*: definition and description of specific parameters, which can be assessed through geophysical investigations (e.g., stiffness, shear modulus, moisture conditions etc.).
4) *Single failure modelling*: reconstruction of single landslide events triggered by earthquakes.
5) *Single failure impact/risk modelling*: analysis focused on the assessment of the effects induced by a single EQTL on human activities (e.g., damaging of buildings).

6) *Single landslide comparison*: analysis focused on the comparison between single landslides triggered by earthquakes.



### 2.1.3 Earthquake information

The 804 identified papers describe ground effects related to 135 earthquakes. For each earthquake we collected a set of data using as main source of information the ANSS Comprehensive Earthquake Catalog (ComCat), implemented by USGS (http://earthquake.usgs.gov/earthquakes/search/). In the catalogue, the available information depends on the date of occurrence. For earthquakes occurred before 1900 (18 events), it was possible to attribute only a broad localization of the epicentre, an estimation of the date of occurrence and the macroseismic intensity, which allows to quantify the shaking level

from observations of the effects on buildings, environment, and people (Masi et al., 2020). For earthquakes that occurred after 1900, it was instead possible to get more detailed information and it was possible to associate to each seismic event the data listed in Table 1: earthquake ID (D1), the name (D2), the time of occurrence (D3) in the format *yy-mm-dd-hh:mm-ss UTC* (Coordinated Universal Time), and the country of occurrence (D4). A distinction was made between single and multiple events (D5): an earthquake was considered as multiple when the seismicity nearby the main shock was characterized by at least 10

events with $M_w$ (moment magnitude) $\geq 5$. In the database we have also indicated the focal mechanism (D6). Specifically, the main deformation style of the earthquake-generating fault was defined using the Kaverina-type DC classification diagram (Kaverina et al., 1996), which considers 7 different fault mechanisms: normal, normal-strike-slip, reverse, reverse-strike-slip, strike-slip, strike-slip-normal, and strike-slip-reverse. From the Global Centroid Moment Tensor Catalogue (GCMTC) (https://www.globalcmt.org/), we retrieved the moment tensor solution (D7) for 96 out of 135 investigated earthquakes. We

used the moment tensor solution as input for the FMC software (Álvarez-Gómez, 2019), which allows to automatically generate a Kaverina-type DC classification diagram. In the cases where the moment tensor solution was missing, we defined the focal mechanism from the specific literature, if available (D8). The different types of magnitude, i.e., $M_L$-Richter local magnitude (D9), $M_s$-Richter surface wave magnitude (D10), $M_w$-moment magnitude (D11), and MCS-Mercalli–Cancani– Sieberg intensity scale (D12), were also added into the database, together with the epicentre coordinates (latitude - D13 and

longitude - D14) and the hypocentre depth (D15). For each record, we specified the source of information (D16). As mentioned above, most of the information derives from the ANSS catalogue but for several historical earthquakes, we used the National Centers for Environmental Information/World Data Service (NCEI/WDS) Global significant Earthquake Database (https://www.ngdc.noaa.gov/hazel/view/hazards/earthquake/search), which includes over 5,700 earthquakes from 2150 BC to the present.

**2.2 Structure and features of the web-GIS database**

   After the compilation of the database, the information of each article and earthquake were implemented within a specific web-GIS (http://194.119.218.119/en/map/EQUILS_LIT_DB/). The web-GIS allows to manage and consult such information according to the geographical data related to it. This means that only those articles dealing with specific earthquake events were included in the web-GIS. For all other articles, the label "Not Applicable" is reported in the fields D1-D4.





The web-GIS was implemented through G3W-SUITE (https://g3wsuite.it/en/g3w-suite-publish-qgis-projects/), which is a
        modular client-server application that fully integrates the QGIS Python API and allows to publish and manage QGIS
        cartographic projects. Basically, it comprises two geographic layers and two tables. The geographical layers refer to the
        earthquakes, which are located with respect to their epicentre (latitude- D13 and longitude-D14), and the countries, which are
        defined based on the "world_boundaries" shapefile (https://public.opendatasoft.com/explore/dataset/world-administrative-
boundaries/export). The two tables ("articles_earthquake" and "articles_country") are linked to the corresponding geographic
        layers through a one-to-many relationship.

        As a first step, we built the "articles_earthquake" table starting from the EQTLs literature database. In this respect, it is
        important to notice that numerous articles included in the database deal with more than one earthquake. For this reason, for
        the construction of the table we repeated the record for each addressed earthquake, then using "Earthquake ID" (D1) as foreign
key of the table. In this way, it was possible to join each earthquake of the "earthquake" layer to many records (i.e., papers) of
        the "articles_earthquake" table. Similarly, we built the "articles_country" table by repeating the record for each country where
        the addressed earthquakes occurred, then using the country (D4 – earthquake country) as foreign key of the table. In this case,
        an inner join was created between each country of the "world_boundaries" geographical layer and many records (i.e., articles)
        of the "articles_country" table.

The web-GIS has a user-friendly interface consisting of different panels. The left panel contains the following fields:

        1)  Metadata: general information regarding the project;

        2)  Charts: two different graphs are reported, i.e., n. articles vs. country and n. articles vs. earthquake;

        3)  Search: a search tool which allows to carry out alphanumeric searches on the available layers (i.e., "earthquake" and
            "world_boundaries");

4)  WMS: where the user can add custom WMS services to the project;

        5)  Map: which contains a structured list of the layers (i.e., "articles_earthquake", "earthquake", "articles_country",
            "world_boundaries") and the corresponding legend. By right clicking on the name of each layer, it is possible to view the
            attribute table.

        In the lower part of the map area there is an information bar showing the display scale, the mouse coordinates, and the
projection/coordinate system. Finally, in the right side of the map area there is a panel with several commands for zooming
        in/out, measuring distances and areas, uploading new layers, and taking snapshots of the map area. In this panel, the user can
        also perform punctual queries (i) by clicking on specific earthquakes or countries, in order to view the main information of
        that specific feature, including the one-to-may relation data. In the latter table, the user can also perform specific searches for
        filtering the results through keywords.




## 3 Preliminary analysis of the literature database

In the following sub-chapters, we present and discuss a preliminary analysis on the main topics and sub-topics (Sect. 3.1) and on the earthquakes addressed and described in the articles (Sect. 3.2). In addition, in Section 3.3 we have evaluated the information on the addressed topics and the listed earthquakes for inferring general and specific aspects of the EQTLs scientific literature.

### 3.1 Analysis of article topic

The preliminary analysis of the articles included in the database allowed to recognize different themes and commonalities, and to group them according to main topics and sub-topics (see Sect. 2.1.2). In many cases, we assigned more than one main topic and/or sub-topic to a single article in relation to the variety of the performed analyses/activities.

As regards the main topics, Figure 3A reveals that the most represented category is "Regional landslide analysis", with more than half (53.6%) of the articles, followed by "Single landslide analysis" (27.4%), "Not applicable" (19.5%) and "Other" (3.5%).

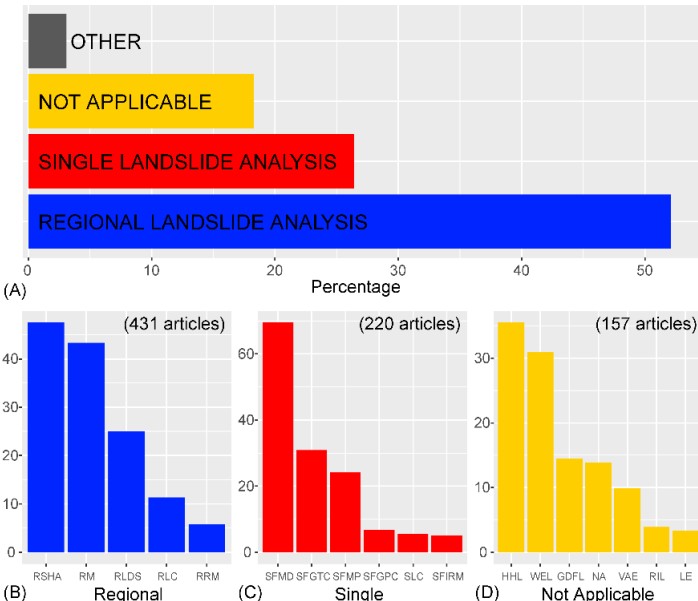

**Figure 3: Percentage of articles with respect to the main topics (A) and sub-topics for "Regional landslide analysis" (B), "Single landslide analysis" (C) and "Not applicable" (D) category. The percentage is calculated with respect to the total number of articles included in the considered category (in brackets). Legend: RSHA: regional susceptibility/hazard assessment, RM: regional mapping, RLDS: regional landslide descriptive statistics, RLC: regional landslide comparison, RRM: regional risk modelling, SFMD: single failure modelling, SFGTC: single failure geotechnical characterization, SFMP: single failure mapping, SFGPC: single failure geophysical characterization, SLC: single landslide comparison, SFIRM: single failure impact/risk modelling, HHL: historical/Holocene landslides, WEL: water environment landslides, GDFL: ground deformation, failure and liquefaction processes, NA: not applicable, VAE: vegetation and EQTLs, RIL: rainfall-induced landslides, LE: landscape evolution.**



### 3.1.1 "Regional landslide analysis" category

The two most numerous sub-categories in "Regional landslide analysis" category (Figure 3B) include articles which discuss "Regional Susceptibility/Hazard Assessment" analyses (RSHA) and "Regional Mapping" activities (RM) (46.9% and 43.4%,
respectively). The third sub-category (Regional Landslide Descriptive Statistics - RLDS) presents statistical descriptions of EQTLs data and contains about a quarter of the articles (25.1%). Two further sub-categories discuss "Regional Landslides Comparison" (RLC) and "Regional Risk Modelling" (RRM) and contain only few articles (11.8% and 5.8%, respectively).

In the sub-category RSHA (202 articles), a large part of the articles uses and describes a modified versions of the Newmark (1965) model (e.g., Jin et al., 2019) or statistical methods such as logistic regression (e.g., Polykretis et al., 2019), artificial
neural network (e.g., Tian et al., 2019), fuzzy logic (e.g., Razifard et al., 2019). In the set of articles that introduce modified versions of the Newmark model, the EQTL scenario related to a specific seismic event usually implies the exceedance of pre-established co-seismic displacement thresholds (Romeo, 2000), while in the group of statistical models, the outcome generally consists in susceptibility maps resulting from the weighting of different environmental factors (e.g., slope angle, geology, vegetation cover, ground shaking intensity). In both cases the analysis of the EQTL phenomenon concerns almost exclusively
the triggering stage, while propagation and deposition processes are seldom examined (e.g., Guo et al., 2014). In addition, no specific methodological variations are presented with respect to the investigated landslide type (e.g., rockfall, earthflows etc.). The high number of papers included in the RM sub-category (187 articles) highlights the importance of landslide inventories for analyses over large areas. In this respect, 79 out of the 187 RM articles can be also classified within the third sub-category RLDS, since they include not only a description of one or more EQTL inventories, but also a statistical analysis of the features
of the mapped landslides (e.g., Roback et al., 2018). Conversely, other 66 RM articles are exclusively focused on mapping procedure, which are performed with different methods and techniques such as: field investigation (e.g., Martino et al., 2017), analysis of aerial imagery (e.g., Saito et al., 2018) and satellite data (e.g., Hu et al., 2019).

As regards the RLC and RRM sub-categories, the relatively low number of articles (51 and 25, respectively) suggests the high specificity of the themes. In the first case, the comparison between landslide inventories can be performed for evaluating
potential differences among earthquakes occurring in the same area (e.g., Jibson et al., 2020) or for inferring general rules (Tanyas and Lombardo, 2020) and/or correlations, e.g., magnitude vs. affected area (e.g., Marc et al., 2017), ground motion vs. landslide size (e.g., Jibson and Tanyas, 2020). In the case of RRM articles, the analyses are focused on specific anthropic elements such as buildings, roads, and railways (e.g., Vega and Hidalgo, 2016) or can be framed within a wider risk assessment (e.g., Martino et al., 2020).

### 240 3.1.2 "Single landslide analysis" category

The largest number of articles (153 out of 220) within the "Single landslide analysis" category (Figure 3C) is included in the "Single Failure Modelling" (SFMD) group. These articles thoroughly investigate the triggering (e.g., Zhang et al., 2018) and/or the propagation process (e.g., Li et al., 2017) of single landslide events induced by specific seismic inputs. Although different types of approaches have been proposed by the authors, e.g., laboratory experimental testing (e.g., Pu et al., 2020), numerical
modelling still represents the most widely used technique. Over the years, numerous types of increasingly sophisticated models





have been developed (Jibson, 2011), which allowed to simulate complex physical processes related to EQTLs, as for example progressive slope failure induced by strain-softening behaviour (e.g., Islam et al., 2019), dynamic fragmentation (e.g., Zhao and Crosta, 2018), and pore-water pressure variation on the sliding surface (Huang et al., 2019).

The SFMD sub-category also comprises 39 out of 68 articles included in the second sub-category ("Single Failure Geotechnical
Characterization"-SFGTC). In fact, numerical modelling often requires a detailed definition of the geotechnical parameters of the landslides body, which can be measured directly on field (e.g., Gratchev and Towhata, 2010) or estimated through laboratory tests and parametric analyses (e.g., Dang et al., 2016). It is important to specify that, in many cases, the authors defined the input parameters of the numerical model simply based on literature values referred to the same or similar landslide event (e.g., Nian et al., 2020).

Similarly, as described for regional scale, "Single Failure MaPping" (SFMP) sub-category includes all those articles dealing with mapping of a single landslide body (e.g., Bozzano et al., 2008). Within this sub-category it is also possible to find almost all the articles belonging to the "Single Failure GeoPhysical Characterization" (SFGPC). This point suggests how geophysical investigations are often associated with the detailed reconstruction of the geometry of the landslide (e.g., Havenith et al., 2002). As regards "Single Landslide Comparison" (SLC) and "Single Failure Impact/Risk Modelling" (SFIRM), we identified a
relatively low number of articles (12 and 11, respectively), as in the case of regional scale analyses. In the first instance, the comparison concerns landslides occurred during the same earthquake (e.g., Nakamura et al., 2014) or in response to other events occurred in different regions (e.g., Aydan, 2016). In SFIRM articles, the effects of an EQTL on human activities can be estimated after a specific event (*ex post* assessment e.g., Cui et al., 2012) or *a priori* through the reconstruction of specific single landslide risk scenarios (e.g., Mousavi et al., 2011).

**3.1.3 "Not applicable" category**

As mentioned above, the category "Not applicable" contains articles which address and discuss themes only partially related to EQTLs. Considering the not negligible number of articles (157), we performed their analysis and evaluation. On the contrary, we decided to disregard the "Other" category, since the low number of articles (28) did not allow us to infer general aspects and significant commonalities.

The examination of the article classified as "Not applicable" consented to identify different sub-categories (Figure 3D). The most numerous ("Historical Holocene Landslides"-HHL) contains 53 articles (33.8%) dealing with landslides occurred in historical times (i.e., before 1900) and, more generally, during the Holocene. In case of historical failures, only generic information about the triggering earthquake is available, while in the second one (landslide occurred during the Holocene), it is often difficult to recognize with certainty the earthquake as triggering factor. In these articles, the authors often use terms
that refer to the possible nature of the landslide trigger, such as: "was probably triggered" (Pérez-López et al., 2019) or "most likely triggered" (Lv et al., 2014). In this group, several authors also describe dating techniques, such as dendrochronological (e.g., Struble et al., 2020), lichenometric (e.g., Pérez-López et al., 2019) and isotopic analyses (e.g., Kojima et al., 2014). For landslide events occurred in historical times, archival records often represent the main source of information (Koukouvelas et al 2020). The back-analysis of such events was also used to assess the intensity and the location of historical earthquakes as





described, for instance, by Rodríguez-Peces et al., 2011 for the 1755 Lisbon (Portugal) and 1884 Arenas del Rey (Spain) earthquakes.

Beyond the historical landslides, we also decided to include in the "Not applicable" category all those articles dealing with landslides related to the geological hazard chain potentially resulting after an earthquake (Fan et al., 2019). In this respect, we identified three sub-categories:

1)   Water Environment Landslides" (WEL): this sub-category comprises 46 articles (29.3%) specifically focused on the secondary effects of EQTLs in water environments, such as landslide-induced tsunamis (e.g., Takagi et al., 2019) and floods related to river landslide dams (e.g., Fan et al., 2012). In case of historical earthquakes, EQTLs in water environments are generally used as evidence for confirming past tsunamis (e.g., Kitamura et al., 2020) and river damming events (e.g., Ehteshami-Moinabadi and Nasiri, 2019) or, more generally, the occurrence of one or multiple earthquakes in

290       a specific area (e.g., Goto et al., 2010).

  2)   Rainfall-Induced Landslides (RIL): the 9 articles (5.7%) included in this sub-category were classified as "Not applicable" since earthquake represents a predisposing factor for subsequent rainfall-induced failures, i.e., post-seismic landslides (Tanyas et al., 2021). In general, these papers del with comprehensive landslide hazard assessments (e.g., Hong and Adler, 2007) or cascading effects' evaluations (e.g., Tunas et al., 2020).

3)   Landscape Evolution (LE): these 8 articles (5.1%) investigate the role of EQTLs as a surface process in the framework of landscape evolution, also considering other morphogenetic processes as, for example, tectonic uplift (e.g., Gallousi et al., 2007; Li et al., 2019) or fluvial sediment discharge (e.g., Hovius et al., 1997; Marc et al., 2016).

Another sub-category of articles classified as "Not applicable" includes works mainly focused on other earthquake-induced ground effects which can occur concurrently with EQTLs. In particular, 22 articles (14.3% of the total) describe and discuss

EQTLs together with co-seismic Ground Deformations and Failure/Liquefaction processes (GDFL) (e.g., He et al., 2020). However, it is important to stress that liquefaction is a relevant and important earthquake-induced ground effect which frequently occurs in flat areas, thus independently of EQTLs. Many scientific articles are specifically focused on this topic, but they are not listed in our database since the word "liquefaction" is not one of the searching keywords.

The analysis of "Vegetation And EQTLs" (VAE) is addressed by 15 articles (9.7%) and represents another sub-topic. These

articles are focused on the stabilising effect of vegetation recovery at landslide sites after earthquake events (Yang et al., 2018) or, conversely, on vegetation change/alteration induced by EQTLs (e.g., Cheng et al., 2012).

Finally, 21 articles were classified as fully "Not Applicable" (NA) since the addressed topic is not related to EQTLs, such as the dynamic response of slopes based on in-situ monitoring systems (Moore et al., 2011; Lenti et al., 2015) or the evaluation of the hydrological response of landslide bodies under seismic loading (Beyabanaki et al., 2016; O'Brien et al., 2016).

**3.2 Analysis of earthquake information**

As mentioned in Sect. 2.1.3, the 804 articles collected in the database discuss and present ground effects of 135 earthquakes, mainly located in Italy, Japan, USA, and China (Table 2). The earthquakes analysed in the articles agrees with the typical





distribution of the main seismicity at the global scale (Figure 4), with most of the events located along the boundaries of the tectonic plates. About 80% of the world's seismicity is situated along the circum-Pacific margin, mainly in correspondence of

subduction zones, where earthquakes are caused by thrust-type and transcurrent-type faults (Toriumi, 2021). The focal mechanisms of the 135 earthquakes are mainly associated to reverse/reverse-strike-slip faults (52 events, which correspond to 38.5% of the total) and, secondly, to faults with a prevalent strike-slip component (44 events, which correspond to 32.6% of the total) (Table 1 and Figure 5). Moreover, the reverse-faulting mechanism characterizes not only most of the identified events, but also the strongest (Mw ≥ 8) ones, in agreement with the general observation for which the largest earthquakes

mainly occur in subduction zones (Funiciello et al., 2020).

As regards the intensity, 46 earthquakes (34.1% of the total) have a $M_w \geq 7$ (Table 3). Although it may seem like a large number, the data collected between 2000 and 2021 (https://www.usgs.gov/programs/earthquake-hazards/lists-maps-and-statistics) indicate that the average yearly number of $M_w \geq 7$ earthquakes in the world is 15. This implies that strong earthquakes occurring worldwide are more numerous than those effectively analysed by the scientific community, at least in the framework

of EQTLs. This can be explained considering that, in principle, the investigated earthquakes are those more impactful for human activities. In the specific case of EQTLs, the scientific interest generally focuses on earthquakes occurring in populated areas or nearby. In this respect, most of the articles included in the database deal with earthquakes occurred in the Asian inland and surrounding areas (Figure 4). The devastating 2008 Wenchuan earthquake ($M_w$ 7.9) is certainly the most studied event (179 articles), while the 1999 Chi-Chi earthquake ($M_w$ 7.7) is the second one, although with a much lower number of articles

(59) (Table 4). The most studied earthquake which did not occur on the Asian continent is the 1994 Northridge earthquake ($M_w$ 6.7) with 13 articles. In general, earthquakes occurred in China are the most studied events in the framework of EQTLs (i.e., 255 articles distributed among 13 earthquakes), followed by those located in Japan (i.e., 78 articles for 16 earthquakes) (Figure 6A-B). It is interesting to note that the earthquakes occurred in Italy, albeit more numerous (i.e., 19 events), are not as studied, since only 39 articles address these events, with a maximum value of 9 articles for the 2016 Amatrice earthquake ($M_w$

6.2) (Figure 6C). This point can be explained considering that earthquakes tend to arouse scientific interest mainly at national level. In this sense, only 2 out of 39 articles do not include Italian authors. As regards the 16 earthquakes occurred in the United States (Figure 6D), only the 1994 Northridge earthquake has been examined in more than 5 articles (Table 4). Thus, if we consider that only 30 articles of the database focus on earthquakes in USA, we can assert that these events, despite the high magnitude (average $M_w$ 7.1), have been analysed to a lesser extent, at least in relation to EQTLs.

Finally, it is interesting to highlight that 18 out of 135 earthquakes have been labelled as historical (i.e., before 1900). Only 20 articles focus on these events, with a maximum value of 4 papers for the 1786 Moxi earthquake (Table 5). Based on what explained in Sect. 3.1.3, these articles have been classified as "Not applicable" since they concern historical landslides. Specifically, 6 articles can be also included in the sub-category "water environment landslides", while only one article, which describes the effects of the 1811-1812 New Madrid earthquakes (Tuttle and Barstow, 1996), can be linked to the "ground

failure-deformation/liquefaction" sub-category.


| Country | Earthquake (# and M) | | Country | Earthquake (# and M) | |
|---|---|---|---|---|---|
| Italy | 19 | | Nepal | 2 | |
| Japan | 16 | | Papua New Guinea | 2 | |
| USA | 16 | | Philippines | 2 | |
| China | 13 | | Portugal | 2 | |
| Iran | 6 | | Russia | 2 | |
| New Zealand | 6 | | Algeria | 1 | |
| Spain | 5 | | Bulgaria | 1 | |
| India | 4 | | Costa Rica | 1 | |
| Perù | 4 | | Georgia | 1 | |
| Taiwan | 4 | | Guatemala | 1 | |
| Canada | 3 | | Haiti | 1 | |
| Ecuador | 3 | | Korea | 1 | |
| Greece | 3 | | Pakistan | 1 | |
| Indonesia | 3 | | Republic of Macedonia | 1 | |
| Chile | 2 | | Tajikistan | 1 | |
| Mexico | 2 | | The Netherlands | 1 | |
| El Salvador | 2 | | Turkey | 1 | |
| Kyrgyzstan | 2 | | | | |

**Table 2: Number of earthquakes listed in the database for each country. The magnitude of each event is reported in the coloured bar. For each country, the lines with the black dot show the first and the last earthquake.**


| Moment magnitude | Counts |
|---|---|
| $4 \leq M_w < 5$ | 2 |
| $5 \leq M_w < 6$ | 17 |
| $6 \leq M_w < 7$ | 48 |
| $7 \leq M_w < 8$ | 36 |
| $8 \leq M_w < 9$ | 8 |
| $M_w \geq 9$ | 2 |

**Table 3: Number of earthquakes identified for different intervals of moment magnitude ($M_w$).**


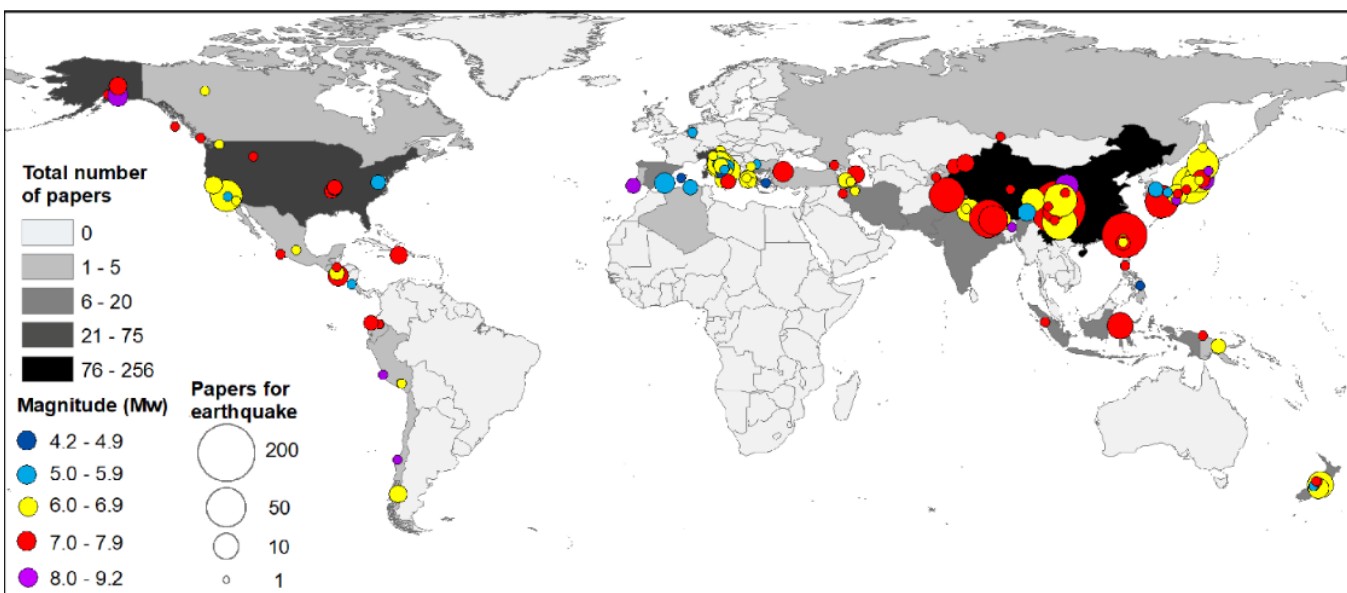

**Figure 4: Spatial distribution of the 135 earthquakes identified in the literature search. Colour indicates the magnitude, while the**
**size of the circle is proportional to the number of articles dealing with each earthquake. Countries are classified based on the number**
**of articles collected in the database.**

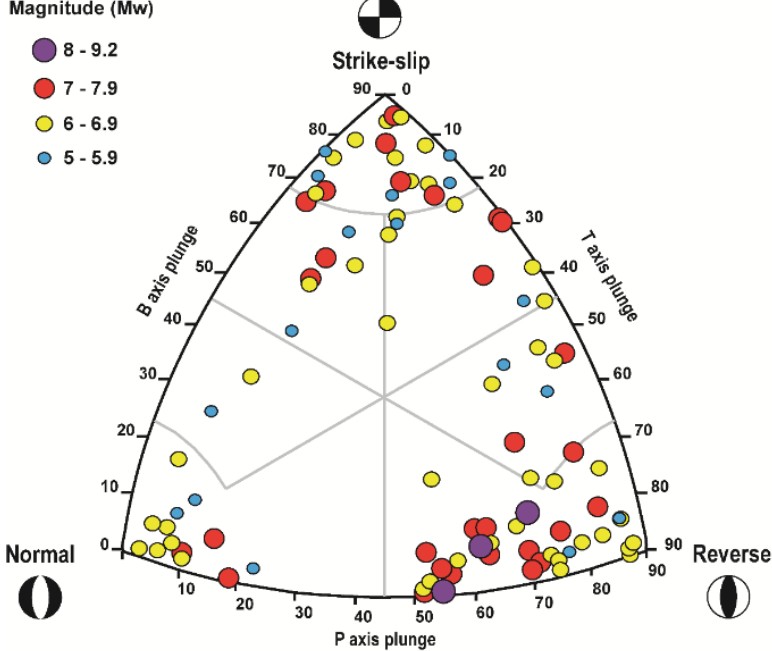

**Figure 5: Definition of the focal mechanism of 96 out of 135 investigated earthquakes according to the Kaverina-type DC**
**classification diagram. Several earthquakes are not plotted due to the lack of information concerning P, T and B centroid moment**
**tensor axes.**



| ID | Earthquake | Country | Earthquake date | $M_w$ | N. articles |
|---|---|---|---|---|---|
| chin_1 | Wenchuan | China | 2008-05-12 06:28:01 UTC | 7.9 | 179 |
| taiw_1 | Chi-Chi | Taiwan | 1999-09-20 17:47:18 UTC | 7.7 | 59 |
| nep_1 | Gorkha | Nepal | 2015-04-25 06:11:25 UTC | 7.8 | 28 |
| jap_1 | Niigata-Chuetsu | Japan | 2004-10-23 08:56:00 UTC | 6.6 | 27 |
| chin_2 | Lushan | China | 2013-04-20 00:02:47 UTC | 6.6 | 23 |
| pak_1 | Kashmir | Pakistan | 2005-10-08 03:50:40 UTC | 7.6 | 19 |
| chin_5 | Wenping | China | 2014-08-03 08:30:13 UTC | 6.2 | 18 |
| chin_8 | Jiuzhaigou County | China | 2017-08-08 13:19:49 UTC | 6.5 | 16 |
| jap_2 | Kumamoto | Japan | 2016-04-15 16:25:06 UTC | 7 | 16 |
| jap_6 | Hokkaido Eastern Iburi | Japan | 2018-09-05 18:07:59 UTC | 6.6 | 13 |
| usa_4 | Northridge | USA | 1994-01-17 12:30:55 UTC | 6.7 | 13 |
| nep_2 | Gorkha_2 | Nepal | 2015-05-12 07:05:19 UTC | 7.3 | 11 |
| jap_4 | Iwate–Miyagi Nairiku | Japan | 2008-06-13 23:43:45 UTC | 6.9 | 10 |
| ita_4 | Amatrice | Italy | 2016-08-24 01:36:32 UTC | 6.2 | 9 |
| indo_1 | Palu | Indonesia | 2018-09-28 10:02:45 UTC | 7.5 | 8 |
| chin_3 | Minxian-Zhangxian | China | 2013-07-21 23:45:56 UTC | 5.9 | 7 |
| ita_6 | Norcia | Italy | 2016-10-30 06:40:18 UTC | 6.6 | 7 |
| nze_1 | Kaikōura | New Zealand | 2016-11-14 00:34:22 UTC | 6.5 | 7 |
| ita_1 | Irpinia | Italy | 1980-11-23 18:34:53 UTC | 6.9 ($M_s$) | 6 |
| chin_4 | Haiyuan | China | 1920-12-16 12:05:55 UTC | 8.3 | 5 |
| chin_6 | Yushu | China | 2010-04-13 23:49:38 UTC | 6.9 | 5 |
| ind_1 | Chamoli | India | 1999-03-28 19:05:11 UTC | 6.6 | 5 |
| ita_5 | Castelsantangelo sul Nera | Italy | 2016-10-26 19:18:08 UTC | 6.1 | 5 |
| ita_12 | Umbria-Marche | Italy | 1997-09-26 09:40:26 UTC | 6 | 5 |

**Table 4: List of the main characteristics of the earthquakes analysed in more than 5 articles.**

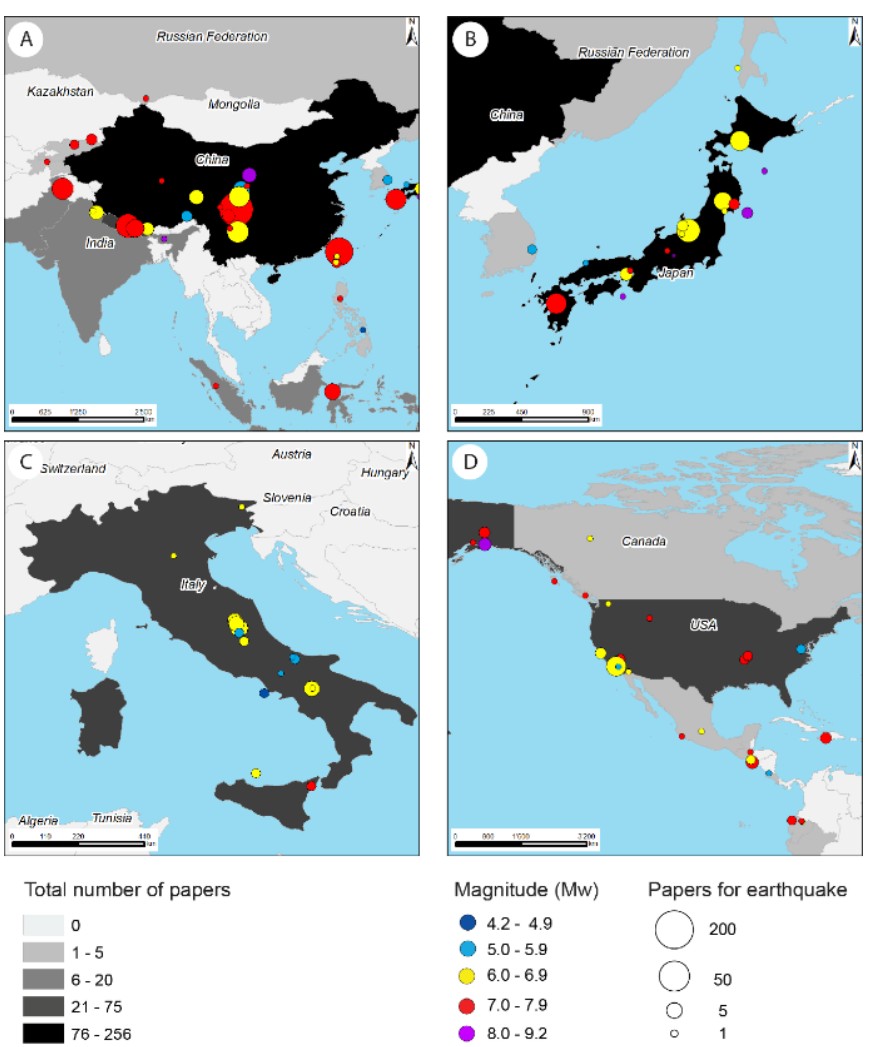

**Figure 6: Spatial distribution of the earthquakes listed in the database for four countries, i.e., China (A), Japan (B), Italy (C), and USA (D). Colour indicates the magnitude, while the size of the circle is proportional to the number of articles dealing with the earthquake. Countries are classified based on the number of articles collected in the database.**



| ID | Earthquake | Country | Earthquake date | $M_s$ | N. articles |
|----|-----------|---------|-----------------|-------|-------------|
| chin_7 | Moxi | China | 1786-06-01 | 7.75 | 4 |
| ita_13 | Calabria | Italy | 1783-02-06 | 5.9 | 2 |
| por_2 | Lisboa | Portugal | 1755-11-01 | 8.5 ($M_w$) | 2 |
| usa_9 | New Madrid 1 | USA | 1811-12-16 | 8.5 | 2 |
| usa_10 | New Madrid 2 | USA | 1812-01-23 | 8.4 | 2 |
| usa_11 | New Madrid 3 | USA | 1812-02-07 | 8.8 | 2 |
| chin_10 | Xichang | China | 1850-12-09 | 7.5 | 1 |
| chin_11 | Tongwei | China | 1718-06-19 | 7.5 | 1 |
| ind_3 | Assam | India | 1897-06-12 | 8.7 | 1 |
| indo_3 | Ambon | Indonesia | 1674-02-17 | 6.8 | 1 |
| ita_16 | offshore Apulia | Italy | 1743-02-20 | 6.9 | 1 |
| ita_17 | offshore Sicily | Italy | 1693-01-09 | 6 | 1 |
| jap_13 | Keichou-Fushimi | Japan | 1596 | 7 | 1 |
| jap_15 | Totomi-Jishin | Japan | 714 | unknown | 1 |
| jap_16 | Hietsu | Japan | 1858-04-09 | 7.45 | 1 |
| jap_9 | Hoei | Japan | 1707-10-28 | 8.4 | 1 |
| spa_2 | Arenas del Rey | Spain | 1884-12-25 | 6.5 ($M_w$) | 1 |
| usa_13 | Lake Chelan | USA | 1872-12-15 | 6.8 ($M_w$) | 1 |

**Table 5: List of historical earthquakes collected in the database.**

### 3.3 Combined analysis of article topics and earthquake information

Based on the above information, we have performed combined analyses to evaluate: i) the addressed main topics and sub-
topics over time and ii) to evaluate the type of scientific evaluation performed for specific earthquake events. Figure 7 reveals
that from the second half of 2000s the "regional landslide analysis" articles have always been more numerous than "single
landslide analysis" ones, with an almost constant proportion during the entire period. This confirms the increasing use, in the
last decades, of remote sensing data, especially satellite imagery ones, for the generation of EQTLs inventories (Fan et al.,
2019). This point, alongside with the development of calculation skills/performances of advanced computers, greatly enhanced
and expedited the analysis of terrain data, especially over large areas.

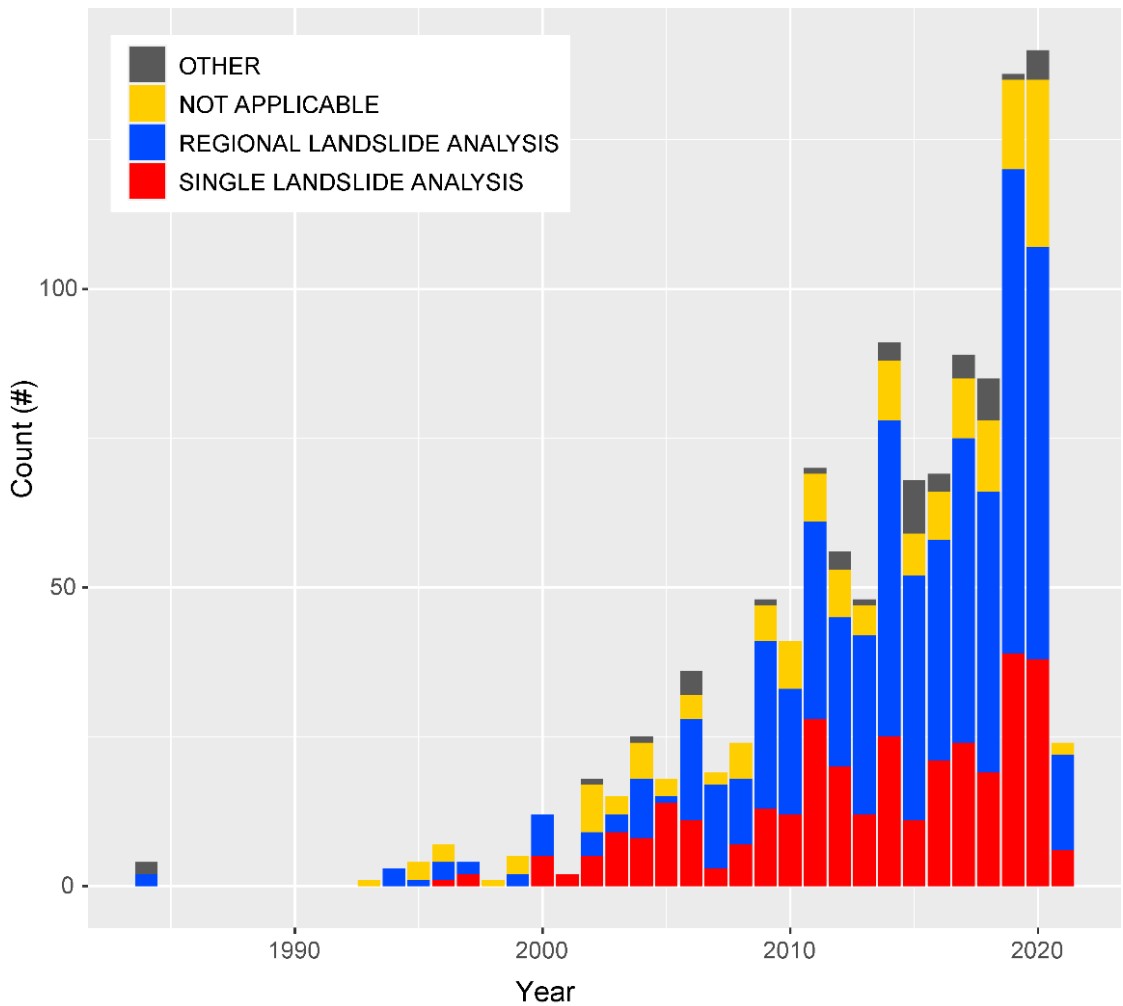

**Figure 7: The graph shows the temporal distribution of the main topics addressed by the authors in the 37-year period from April 1984 to February 2021.**

As regards the analysis of specific earthquakes events, we selected the first five earthquakes listed in Table 4 (i.e., Chi-Chi 1999, Niigata-Chuetsu 2004, Wenchuan 2008, Lushan 2013 and Gorkha 2015), for which more than twenty articles were published. The first three occurred more than ten years ago and can be considered suitable for evaluating potential topic trends over the years.

     In terms of main topics, in the years immediately after the Chi-Chi earthquake, only "single landslide analysis" articles were

published (Figure 8). On the contrary, in the first years after Niigata-Chuetsu and Wenchuan earthquakes, also "regional landslide analysis" studies were performed. For Wenchuan earthquake, the number of "regional landslide analysis" articles are even greater than that of "single landslide" ones until 2014, i.e., six years after the event. The increasing trend of "regional landslide analysis" articles is furthermore confirmed by the Lushan 2013 and Gorkha 2015 events. This result suggests that, in





the recent years, researchers have performed regional analyses as a priority with respect to slope-scale studies. Such a preference may be put in relation with the increasing availability of remote sensing data and high-performance computational tools, which enhance the execution of regional-scale analyses in a short time. On the contrary, the collection of data generally employed for slope-scale analyses can be extremely time-consuming, thus affecting the timing of publication. Within "regional" articles we also noticed that, in general, "mapping" (RM) is the prevailing activity in the first period after the event, while "modelling" (RSHA) tends to increase over time. RM articles can still be published also several years after the event:

these works can include updates of existing inventories (e.g., Chen et al., 2020) or *ex-novo* mapping for in-depth analyses of landslide activity (e.g., Liu et al., 2020).

As regards "single landslide analysis" articles, "modelling" (SFMD) is the prevailing activity and shows an increasing trend over the years (Figure 8). Alongside "modelling", "geotechnical characterization" (SFGTC) and "mapping" (SFM) are generally carried out, especially in the first years after the Chi-Chi and Niigata-Chuetsu earthquakes, while other types of sub-

topics are substantially secondary. In the case of Wenchuan, different SFM articles have been published also in recent years (e.g., Cui et al., 2017). In this respect, although more than 60,000 landslides have been triggered during the event (Gorum et al., 2011), only few failures have been investigated individually. This leaves room for the verification to the real triggering and conditioning mechanisms, which as a matter of fact rely only on a few studies. Among these, the Daguangbao landslide is the most studied (e.g., Luo et al., 2020), probably due to its great size ($7.5 \times 10^8$ m$^3$). As regards Lushan and Gorkha event,

very few articles have been published in the framework of "single landslide analysis" topic; thus, it was not possible to observe any specific trend.






**Figure 8: Temporal analysis of the main topics (first row), "regional" (second row) and "single landslide" sub-topics (third row)**
**addressed by the articles dealing with Chi-Chi 1999 (first column), Niigata-Chuetsu 2004 (second column), Wenchuan 2008 (third column), Lushan 2013 (fourth column), and Gorkha 2015 earthquake (fifth column)**



## 4 Final remarks and conclusions

In this work, we present a comprehensive database of the main scientific articles published in the last four decades on the EQTLs theme. After the identification of 804 papers, the database was compiled in a GIS environment specifying, for each article, different types of information. Great effort was dedicated to the identification of the addressed earthquakes and to the articles' grouping with respect to main topics and sub-topics relevant to EQTLs. In this sense, we highlighted that more than 50% of the articles focus on regional analyses which, in turn, are equally distributed among "modelling" and "mapping" sub-

topics. On the contrary, "single landslide analysis" works, which represent just over a quarter of the total number of articles, deal more with modelling-type activities and, secondarily, with geotechnical characterization of the landslide body. As regards the addressed earthquakes, although great part of the 135 identified events occurred in four different countries (Italy, Japan, United States and China), earthquakes which took place in China are by far the most studied events (255 articles), even if 179 out of these 255 articles analyse the 2008 Wenchuan earthquake. Finally, by analysing the addressed topics over time for the

five main earthquakes (Chi-Chi 1999, Niigata-Chuetsu 2004, Wenchuan 2008, Lushan 2013 and Gorkha 2015), we pointed out an increasing trend of "regional landslide analysis" articles in the last years, alongside with a growing use of modelling approaches both for regional and single landslide-scale analyses.

However, it is important to stress that the considerations presented in this work just represent general inferences resulting from the analysis of the collected articles. More specific observations would require an in-depth, critical analysis of the articles,

which is beyond the scope of the present study. In fact, this work can be considered as a starting point for further analysis and investigations and is proposed to the expertise and a general reader interested in a broader view of EQTLs

In conclusion, the here-presented web-GIS database can represent a powerful tool for performing cross-correlated literature searches focused on specific main topics/sub-topics in relation to given earthquake events and/or study areas. Furthermore, considering its characteristics, the database can be updated very promptly. In this respect, the authors invite all the readers to

report not only new-published articles on EQTLs theme, but also articles that were published in the past and were not included in the database due to shortcomings in the use of keywords during the literature search on Web of Science database.

## Acknowledgements

This article was prepared in the framework of the activities of the FRA.SI project (multi-scale integrated methodologies for seismically-induced landslides hazard zonation in Italy) funded by the Italian Ministry of Ecological Transition (*Ministero*

*della Transizione Ecologica*).



**Authors contribution**

**Luca Schilirò:** conceptualization, data curation, writing-original draft **Mauro Rossi:** conceptualization, data curation, software, writing-review and editing **Federica Polpetta:** data curation, writing-review and editing **Federica Fiorucci:** web-GIS conceptualization and set-up, data curation, writing-review and editing **Carolina Fortunato:** data curation, writing-review and editing **Paola Reichenbach:** conceptualization, data curation, writing-review and editing, supervision, project administration

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
