# Peer review of "A web-GIS database of the scientific articles on earthquake-triggered landslides"

_Natural Hazards and Earth System Sciences, 2022_

## Community Comment (CC1)

The ISC Event Bibliography (www.isc.ac.uk/event_bibliography/index.php, see also Di Giacomo et al., SRL2014) has the potential to be a useful resource for this study. Launched in 2014 by the International Seismological Centre (ISC, www.isc.ac.uk), this service allows users to search for publications studying seismic events (earthquakes as well other types of seismic sources, e.g., explosion, mine collapses etc.) included in the ISC Bulletin (www.isc.ac.uk/iscbulletin/search/). As such, the ISC Event Bibliography covers the period of instrumental seismology (i.e., seismic events occurring since the beginning of the last century) and with references that go back to 1904. Being associated to seismic event information in the ISC Bulletin, searches are based not only bibliographic parameters (journal, year, author), but also on seismic event information (e.g., origin time, coordinates).

In our database we include works from over 500 distinct journal titles and, to a lesser extent, reports from various institutions (e.g. reports from the USGS Publication warehouse).

The journals we follow are focused on multiple geoscience and engineering themes, and journals specialized in earthquake-triggered landslides (EQTLs) are included (e.g., *Landslides*, *Eng. Geol.*, *Geomorphology* and others as shown in Figure 2).

With this note we want to simply inform the authors that the work they presented here can greatly benefit from the ISC Event Bibliography. Indeed, as the authors put it in the Conclusions ("...the authors invite all the readers to report not only new-published articles on EQTLs theme, but also articles that were published in the past and were not included in the database due to shortcomings in the use of keywords during the literature search on Web of Science database."), we are well aware of the enormous difficulties in creating such a database (we indeed invite authors to notify as for missing references). In this spirit we invite the authors to look into our service and use it as a starting point (surely we cannot help in performing the fine and detailed classification of article topic) for expanding the bibliographic record in their database. Just to give a couple of examples, from the search page the authors can look to all events between 1904 and 2022 associated to papers published in the journals "*Landslides*" and "*Eng. Geol.*" at the following links:

www.isc.ac.uk/cgi-bin/bibsearch.pl?searchshape=POLY&coordvals=&start_year=1904&start_month=01&start_day=01&stime=00%3A00%3A00&end_year=2023&end_month=01&end_day=01&etime=00%3A00%3A00&minyear=1984&maxyear=&sortby=day&publisher=Landslides&authors=

http://www.isc.ac.uk/cgi-bin/bibsearch.pl?searchshape=POLY&coordvals=&start_year=1904&start_month=01&start_day=01&stime=00%3A00%3A00&end_year=2023&end_month=01&end_day=01&etime=00%3A00%3A00&minyear=1984&maxyear=&sortby=day&publisher=Engng+Geol.&authors=

Of course other types of searches can be performed (e.g. based on origin time of the event, see for example work for 2018 Eastern Iburi earthquake, that notoriously generated lots of landslides www.isc.ac.uk/cgi-bin/FormatBibprint.pl?evid=612697604).

The other advantage of the ISC Event Bibliography is that references are associated to the event in the ISC Bulletin, regarded as the definitive and most comprehensive record of the Earth's seismicity. As such the authors would have an easier way to maintain their database as the association to the corresponding EQTL is already provided thanks to our work.

We usually update the ISC Event Bibliography database once a month by adding newly published works, but we never stop to look for past references missing in our system.

In this respect, we are grateful to the authors for the references they collected that allowed us to add about 150 missing works in our system, particularly from journals that rarely publish works related to seismology.

Hope you find this note and our service useful and please do not hesitate to contact me for further discussions.

Domenico Di Giacomo

---

## Author Response (AR1)

**Comment CC1**

The ISC Event Bibliography (www.isc.ac.uk/event_bibliography/index.php, see also Di Giacomo et al., 2014) has the potential to be a useful resource for this study. Launched in 2014 by the International Seismological Centre (ISC, www.isc.ac.uk), this service allows users to search for publications studying seismic events (earthquakes as well other types of seismic sources, e.g., explosion, mine collapses etc.) included in the ISC Bulletin (www.isc.ac.uk/iscbulletin/search/). As such, the ISC Event Bibliography covers the period of instrumental seismology (i.e., seismic events occurring since the beginning of the last century) and with references that go back to 1904. Being associated to seismic event information in the ISC Bulletin, searches are based not only bibliographic parameters (journal, year, author), but also on seismic event information (e.g., origin time, coordinates). In our database we include works from over 500 distinct journal titles and, to a lesser extent, reports from various institutions (e.g., reports from the USGS Publication warehouse). The journals we follow are focused on multiple geoscience and engineering themes, and journals specialized in earthquake-triggered landslides (EQTLs) are included (e.g., Landslides, Eng. Geol., Geomorphology and others as shown in Figure 2). With this note we want to simply inform the authors that the work they presented here can greatly benefit from the ISC Event Bibliography. Indeed, as the authors put it in the Conclusions ("...the authors invite all the readers to report not only new-published articles on EQTLs theme, but also articles that were published in the past and were not included in the database due to shortcomings in the use of keywords during the literature search on Web of Science database."), we are well aware of the enormous difficulties in creating such a database (we indeed invite authors to notify as for missing references). In this spirit we invite the authors to look into our service and use it as a starting point (surely we cannot help in performing the fine and detailed classification of article topic) for expanding the bibliographic record in their database. Just to give a couple of examples, from the search page the authors can look to all events between 1904 and 2022 associated to papers published in the journals "Landslides" and "Eng. Geol." at the following links:

www.isc.ac.uk/cgi-bin/bibsearch.pl?searchshape=POLY&coordvals=&start_year=1904&start_month=01&start_day=01&stime=00%3A00%3A00&end_year=2023&end_month=01&end_day=01&etime=00%3A00%3A00&minyear=1984&maxyear=&sortby=day&publisher=Landslides&authors=

http://www.isc.ac.uk/cgi-bin/bibsearch.pl?searchshape=POLY&coordvals=&start_year=1904&start_month=01&start_day=01&stime=00%3A00%3A00&end_year=2023&end_month=01&end_day=01&etime=00%3A00%3A00&minyear=1984&maxyear=&sortby=day&publisher=Engng+Geol.&authors=

Of course other types of searches can be performed (e.g. based on origin time of the event, see for example work for 2018 Eastern Iburi earthquake, that notoriously generated lots of landslides www.isc.ac.uk/cgi-bin/FormatBibprint.pl?evid=612697604). The other advantage of the ISC Event Bibliography is that references are associated to the event in the ISC Bulletin, regarded as the definitive and most comprehensive record of the Earth's seismicity. As such the authors would have an easier way to maintain their database as the association to the corresponding EQTL is already provided thanks to our work. We usually update the ISC Event Bibliography database once a month by adding newly published works, but we never stop to look for past references missing in our system. In this respect, we are grateful to the authors for the references they collected that allowed us to add about 150 missing works in our system, particularly from journals that rarely publish works

related to seismology. Hope you find this note and our service useful and please do not hesitate to contact me for further discussions.

Domenico Di Giacomo

ANSWER: thank you very much for your comment. In the revised version of the manuscript, we mentioned the ISC Event Bibliography in Section 2as follows:

"In this respect, other sources of information are freely available online, such as the ISC Event Bibliography (http://www.isc.ac.uk/event_bibliography/index.php), which is a database containing publications about seismology, but also other types of topics related to specific earthquakes, including landslides (Di Giacomo et al., 2014)".

Furthermore, as specified in the "Conclusions", the ISC Event Bibliography can certainly represent a useful tool for integrating future versions of our database, As you pointed out, the ISC database could help us in adding articles which are not currently included in our database, likely due to shortcomings in the use of keywords during the literature search on Web of Science database. However, it is also important to notice that the current version of our database includes only articles published on peer-reviewed journals, leaving out other types of documents (e.g., conference proceedings, technical reports). Thus, we believe that the presence of scientific reports within the ISC Event Bibliography could help us even more in expanding the information related to EQTLs for specific, significant earthquake events.

**Comment RC1**

Dear Authors,

I read with great interest your paper and I'm impressed by the amount of data you collected.

The paper is well-written, easy to follow and text is coherent with sections and sub-sections title.

I just have one concern regarding references to the ESI-scale. Landslides are among the secondary effects used to estimate earthquake intensity by means of the ESI scale (Michetti et al., 2007), but I didn't find any reference to the ESI scale in the main text. Furthermore, by exploring the web-gis you developed I just found 11 papers related to the ESI scale, which is a very little number if compared to the 804 papers you collected. Making a search in Google Scholar, using the terms "landslide esi scale", I found papers that are not included in your datasets, or at least I didn't find them by means of the "author" toolbar of the web-gis. Some of these papers are:

- Ferrario et al., 2019 (https://doi.org/10.1007/s11069-019-03718-w)

. Ferrario et al., 2022 (https://doi.org/10.5194/nhess-22-3527-2022)

- Huayong et al., 2019 (https://doi.org/10.1016/j.enggeo.2019.105149)

- Esposito et al., 2013 (https://doi.org/10.1007/978-3-642-31427-8_38)

I understand that adding references to the ESI scale may be time consuming, so my suggestion could be integrated in a version 2.0 of your database.

Anyway, I wish to thank the authors for the impressive work they carried out that could be very useful for the scientific community.

ANSWER: Thank you very much for your comment. According to the suggestion made by the reviewer, in the new version of the manuscript (section 3.1.1), a reference to the ESI scale was added in relation to the Earthquake Environmental Effects Catalogue (EEE) as follows:

"In this respect, it is important to point out the presence of different online catalogues which report information on EQTLs for recent and historical earthquakes. At global scale, it is worth mentioning the "Open Repository of Earthquake-triggered Ground Failure Inventories" (Tanyas et al., 2017), which reports EQTLs and liquefaction effects data for 363 earthquakes, or the "Earthquake Environmental Effects Catalogue" (Guerrieri et al., 2015), which collects information about environmental effects triggered by specific seismic events, whose intensity is expressed according to the ESI (Environmental Seismic Intensity) scale (Michetti et al., 2007)".

As regards the possible absence of several papers related to the ESI scale in our database, we believe that it may be explained with shortcomings in the use of keywords during the literature search on Web of Science database. However, as specified in the "Conclusions" section, this is just a first version of the database, that we are looking to improve through the integration with other sources of information (e.g., ISC Event Bibliography) and with the contribution of other researchers in reporting previous and new articles on EQTLs theme. As regards the four articles mentioned by the reviewer, we would like to specify that:

1) Ferrario et al., 2019 (https://doi.org/10.1007/s11069-019-03718-w) was already included in the database (ID 111);
2) Ferrario et al., 2022 (https://doi.org/10.5194/nhess-22-3527-2022) was not included since it was published in 2022, while our database contains articles published from April 1984 to February 2021, as specified in section 2;
3) Huayong et al., 2019 (https://doi.org/10.1016/j.enggeo.2019.105149) was added in the revised version of the database (ID 805)
4) Esposito et al., 2013 (https://doi.org/10.1007/978-3-642-31427-8_38) was not included since our database currently includes only articles published on peer-reviewed journals, leaving out other types of documents, such as book chapters, as in this case.

**Comment RC2**

Dear Authors,

The present paper includes 804 published works organized into well-selected sections and sub-sections. I read it with interest and consider this database will be a useful tool for researchers on the issue of earthquake-triggered landslides (EQTL).

Overall, I have the following suggestion regarding the Greek earthquakes. A more detailed incorporation of Greek earthquake literature will be particularly interesting for the following reason. Greek earthquakes are commonly related to landslides and rock falls. Also, earthquake activity in Greece is of short recurrence interval, and the data inserted in this database will help researchers in the future. Of course, the data basis of Greek EQTL compared to the 804 papers you collected is small, but consider that the issue of EQTL in Greece is underestimated. Looking in my files, I found some published works that are not included in your datasets, or I didn't find them using the "author" toolbar. These papers are:

- Papadopoulos G.A., Plessa A., 2000. Magnitude–distance relations for earthquake-induced landslides in Greece. Engineering Geology, 58, 377-386.

- Koukouvelas I.K., Mpresiakas A., Sokos E., Doutsos T. 1996. The tectonic setting and earthquake hazards of the 1993 Pyrgos earthquake, Peloponnese, Greece. Journal of the Geological Society London, 153, 39-49.

- Koukouvelas, I.K., Litoseliti, A., Nikolakopoulos K., Zygouri, V.2015. Earthquake triggered rock falls and their role in the development of a rock slope: The case of Skolis Mountain, Greece. Engineering Geology 191, 71–85.

- Zygouri, V., Koukouvelas, I.K., 2018. Landslides and natural dams in the Krathis River, north Peloponnese, Greece. Bulletin of Engineering Geology and the Environment https://doi.org/10.1007/s10064-017-1225-y

- Litoseliti, A., Koukouvelas, I.K., Nikolakopoulos, K.G., Zygouri, V., 2020. An event-based inventory approach in landslide hazard assessment: the case of the Skolis Mountain, Northwest Peloponnese, Greece. ISPRS Int. J. Geo-Inf. 2020, 9(7), 457; https://doi.org/10.3390/ijgi9070457

- Koukouvelas, I.K., Nikolakopoulos, K., Zygouri, V., Kyriou, A., 2020. Post-seismic monitoring of cliff mass wasting using an unmanned aerial vehicle and field data at Egremni, Lefkada Island, Greece. Geomorphology, 367, 107306 doi.org/10.1016/j.geomorph.2020.107306

Considering your paper as it stands, it is complete, and I hope that it will followed by the next version adding more references. Finally, the work done is quite important and helps understand the role of EQTL in the hazard assessment of modern society.

ANSWER: Thank you very much for your comment. As mentioned in answer to Reviewer 1, we are looking to integrate the following versions of the database with other sources of information (e.g., ISC Event Bibliography) and with the contribution of other researchers in reporting previous and new articles on EQTLs theme. In this respect, we are aware that a certain number of articles on EQTLs were not included in the database, likely due to shortcomings in the use of keywords during the literature search on Web of Science database. Thus, we would like to thank the reviewer for letting us know the existence of further, published works concerning EQTLs in Greece, which have been added to the revised version of the database (ID806, 807, 808, 809 and 810). However, we would like to specify that only the article:

- Papadopoulos G.A., Plessa A., 2000. Magnitude–distance relations for earthquake-induced landslides in Greece. Engineering Geology, 58, 377-386

was not added to the database, since it was already included in the preceding version (ID 240).